# Ferroptosis Inducers Kill Mesenchymal Stem Cells Affected by Neuroblastoma

**DOI:** 10.3390/cancers15041301

**Published:** 2023-02-18

**Authors:** Xiangze Li, Qi Wang, Chencheng Xu, Lei Zhang, Jiquan Zhou, Jingchun Lv, Min Xu, Dapeng Jiang

**Affiliations:** Department of General Surgery, Shanghai Children’s Medical Center, Shanghai Jiao Tong University School of Medicine, Shanghai 200127, China

**Keywords:** neuroblastoma, BMSCs, ferroptosis

## Abstract

**Simple Summary:**

Bone marrow metastasis represents poor prognosis in neuroblastoma, and bone marrow mesenchymal stem cells play an important role in this progression. There is no effective way to intervene in bone marrow mesenchymal stem cells. We attempt to find one way to effectively kill these cells. We found that ferroptosis induction might be an recommended approach. Our findings provide new strategies for therapy of neuroblastoma patients with bone marrow metastasis.

**Abstract:**

Bone marrow (BM) is the most common site of neuroblastoma (NB) metastasis, and its involvement represents poor patient prognosis. In accordance with the “seed and soil” theory of tumor metastasis, BM provides a favorable environment for NB metastasis while bone marrow mesenchymal stem cells (BMSCs) have been recognized as a central part of tumor stroma formation. Yet, there is currently no effective method for intervening these BMSCs. We found that BMSCs affected by NB (NB-BMSCs) could significantly promote NB growth and migration. Additionally, tumor cell-endowed BMSCs showed stronger resistance to several chemotherapeutic agents. Surprisingly, NB-BMSCs were more sensitive to ferroptosis than normal BMSCs. NB-BMSCs had lower levels of intracellular free iron while synthesizing more iron-sulfur clusters and heme. Moreover, the Xc^−^/glutathione/glutathione peroxidase 4 (Xc^−^/GSH/GPX4) pathway of the anti-ferroptosis system was significantly downregulated. Accordingly, ferroptosis inducers erastin and RAS-selective lethal 3 (RSL3) could significantly kill NB-BMSCs with limited effects on normal BMSCs. BMSCs from NB patients with BM metastasis also showed poor anti-ferroptosis ability compared with those from NB patients without BM metastasis. In vivo studies suggested that co-injection of mice with BMSCs and NB cells could significantly promote the growth of tumor tissues compared with injecting NB cells alone. However, treatment with erastin or RSL3 resulted in the opposite effect to some extent. Our results revealed that NB-BMSCs were vulnerable to ferroptosis from downregulation of the Xc^−^/GSH/GPX4 pathway. Ferroptosis inducers could effectively kill NB-BMSCs, but not normal BMSCs. This study provides possible new ideas for the treatment of tumor-associated BMSCs in NB patients.

## 1. Introduction

Neuroblastoma (NB) arises from neuroepithelial cells and is the most common solid tumor in children. This disease has an insidious onset with rapid progression [1]. Metastasis has often already occurred by the time of diagnosis, with bone marrow (BM) being the major site of NB metastasis [2]. Recent research has shown that patients with BM metastasis have worse overall survival and event-free survival rates [3,4,5]. The BM microenvironment, known as the “metastatic niche”, provides tumor cells with a suitable residence [6,7]. Although emerging NB treatment strategies are improving therapeutic outcomes, patients with BM involvement still have poor prognoses and few remedies [2].

Bone marrow mesenchymal stem cells (BMSCs) are one of the most important cell subsets in BM. BMSCs can convert to a tumor-promoting phenotype upon tumor induction [8,9,10]. Recent studies have shown that bidirectional regulation occurs between NB cells and BMSCs [11,12,13]. Specifically, NB cell lines could promote osteogenic differentiation of BMSCs to form a favorable pre-metastatic microenvironment conducive to tumor cell growth in BM. BMSCs isolated from BM of NB patients could exert pro-tumorigenic effects in tumor cells. Therefore, therapeutical intervention of BMSCs affected by NB (NB-BMSCs) is extremely important.

Furthermore, NB-BMSCs are not sensitive to common chemotherapy drugs, such as cisplatin, etoposide, vincristine, or doxorubicin [14,15,16,17]. Platinum-based drugs cannot induce BMSC death within the microenvironment, but rather promote BMSCs to produce a variety of cytokines that can benefit tumor growth and drug resistance [14]. The above results suggest that use of conventional therapies could lead to the activation of BMSCs, resulting in poor therapeutic effects.

Ferroptosis is a novel type of cell death that is distinct from other common cell death processes, such as apoptosis, autophagy, and necrosis [18]. Ferroptosis is triggered by iron-dependent lipid peroxidation, which can be induced by increased Fe^2+^ levels and decreased activity of the anti-ferroptosis system [19]. The classical Xc^−^/glutathione/glutathione peroxidase 4 (Xc^−^/GSH/GPX4) pathway is the most important component of the anti-ferroptosis system [20,21,22]. It has been demonstrated that cystine transported into cells through the cystine/glutamate anti-transporter (systemXc^−^) is reduced to cysteine, which can promote glutathione (GSH) production. GSH acts as a cofactor of GPX4 to promote the intracellular reduction of lethal lipid peroxide into nontoxic lipid alcohol. Erastin and RAS-selective lethal 3 (RSL3), as potent ferroptosis-triggering agents, can inhibit the activities of Xc^−^ system and GPX4, respectively [23,24]. Because few studies have investigated the relevance of ferroptosis and tumor-associated BMSCs, a further understanding of the therapeutic potential of ferroptosis in tumor-associated BMSCs of NB is necessary.

In this study, we found that NB-BMSCs were susceptible to ferroptosis, where the Xc^−^/GSH/GPX4 pathway was downregulated markedly. We explored the application of iron in NB-BMSCs, finding that NB-BMSCs had less free iron and ferritin, while synthesizing more heme and iron-sulfur clusters (ISCs). RSL3 or erastin treatment significantly induced NB-BMSC death compared with normal BMSCs, resulting in decreased cell viability and accumulation of lipid peroxide products. We also examined Xc^−^ and GPX4 levels in BM of NB patients and found that both were lower in patients with bone metastasis. Thus, we demonstrate a method that can effectively kill NB-BMSCs with little effect on normal BMSCs. Our findings may provide new strategies for chemotherapy of NB patients with BM metastasis.

## 2. Materials and Methods

### 2.1. Cell Culture

Human NB cell lines SK-N-BE(2), SH-SY5Y, and IMR-32 were purchased from the Chinese Academy of Sciences Cell Bank (Shanghai, China). SK-N-BE(2) and SH-SY5Y cells were cultured in DMEM/F12 medium (Gibco, Billings, MT, USA). IMR32 cells were maintained in high glucose DMEM (Gibco). Human BMSCs were purchased from Cyagen Biotechnology (Suzhou, China) and cultured in Oricell Basal Medium (Cyagen Biotechnology). All culture media were supplemented with 10% fetal bovine serum (FBS, Gibco) and 1% penicillin-streptomycin (Gibco). Cancer-associated fibroblasts (CAF) have been suggested to originate from MSCs [11]. Since these CAFs have similar characteristics with MSCs, we named these CAFs as CAF-MSCs. We obtained CAF-MSCs from fresh NB tumor specimens from patients who underwent surgery at Shanghai Children’s Medical Center as described previously. All of the experiments were approved by the Ethics Committee, Shanghai Children’s Medical Center, Shanghai Jiao Tong University School of Medicine (China) [No.SCMCIRB-Y2019011]. Tumor tissues were washed with PBS, cut into small pieces and digested with 3 mg/mL collagenase I (Sigma-Aldrich, St. Louis, MO, USA) in PBS for 2 h at 37 °C. Cells were then passed through a 70 μm strainer filter and negatively selected for GD2 expression. The collected cells were then plated in 100 mm dishes pre-coated with fibronectin (1 μg/mL; MCE, South Brunswick Township, NJ, USA) and collagen I (3 μg/mL; Sigma-Aldrich) and cultured in DMEM supplemented with 20% FBS and 1% penicillin/streptomycin. The medium was changed every 3 days. After 12 days, cells were collected and subcultured.

### 2.2. Conditioned Medium Preparation

To generate conditioned media (CM), 4 × 10^6^ NB cells were plated in a 100 mm dish in 10 mL culture medium. After 48 h, the culture supernatant of NB cells was collected and centrifuged at 1000× *g* for 10 min at 4 °C. The CM was passed through a sterile 0.45 µm millipore filter before use in experiments [25]. CM was diluted 1:1 with culture medium. We used CM of different NB cells to culture BMSCs. BMSCs cultured with CM of SK-N-BE(2), IMR-32, and SH-SY5Y cells were named NB-BMSC1, NB-BMSC2, and NB-BMSC3 cells, respectively. We used the same method to collect CM of BMSCs and NB-BMSCs.

### 2.3. Cell Migration Assays

A cell migration assay was carried out as described previously [26]. Transwell culture plates with inserts (Corning, Corning, NY, USA) were used for the cell migration assay. NB cells were seeded in the top. CM from BMSCs and NB-BMSCs collected from overnight fresh culture medium was diluted 1:1 with NB cell culture medium containing 15% FBS and applied to the bottom chamber, followed by incubation overnight at 37 °C with 5% CO_2_. Then, the upper chamber was washed three times with PBS before fixing the cells with 4% paraformaldehyde (Servicebio, Wuhan, China). Both chambers were stained with crystal violet (Sigma-Aldrich) and observed under a microscope. We also used a scratch assay to examine NB cell migration. We scratched the middle of a six-well culture plate and cultured the cells with regular medium containing 1% FBS with or without CM. In this experiment, CM was collected from BMSCs and NB-BMSCs fresh culture medium containing 1% FBS. After 24 h, the scratch was observed under the microscope.

### 2.4. Flow Cytometry

NB cell and BMSC apoptosis was determined by annexin V-fluorescein isothiocyanate (FITC)/propidium iodide double staining. Harvested cells (1 × 10^5^ cells) were suspended in 1× binding buffer (Beyotime, Shanghai, China) and stained with FITC-conjugated annexin V and propidium iodide. Then, the cells were analyzed by flow cytometry (FACSCanto™ BD, San Jose, CA, USA). Expression of surface markers on BMSC lines and MSCs isolated from patients were also analyzed. Antibodies against markers used to analyze BMSCs were purchased from BD: FITC Mouse Anti-Human CD45(560976), PE-Cy7 Mouse Anti-Human CD34(560710), APC Mouse Anti-Human CD105(562408), PE Mouse Anti-Human CD73(550257), BV421 Mouse Anti-Human CD90(562556), and Fixable Viability Stain 780(565388). Lipid oxidation was detected by C11-BODIPY^®^ 581/591 (Abclone, Wuhan, China) [27]. C11-BODIPY was applied to cells in-medium at 37 °C for 60 min. The cells were washed with PBS for three times before collection and then analyzed by flow cytometry.

### 2.5. Colony Formation Assay

Cells (1000 per well) in the logarithmic growth phase were seeded in a 12-well culture plates. The culture medium was changed every 3 days. After 14 days, colonies contained 40–50 cells. Each well was washed three times with PBS before fixation with 4% paraformaldehyde. Cells were stained with crystal violet and imaged with a camera.

### 2.6. Heme and Iron Assays

A Heme Assay Kit (Sigma-Aldrich) was used to measure the heme content. In brief, we used ultrapure water to prepare samples and standards. BMSCs and NB-BMSCs were lysed with RIPA lysis buffer (Applygen, Beijing, China) containing a PMSF Cocktail (50×) (Applygen) and centrifuged, followed by the heme assay. Absorbance was measured at 400 nm. Intracellular iron levels were assessed using an Iron Content Assay Kit (Biosharp, Beijing, China). A total of 5 × 10^6^ BMSCs or NB-BMSCs were lysed by ultrasonication and centrifuged at 10,000× *g*. The supernatant was collected for detection and absorbance was compared with the standard.

### 2.7. Cell Viability Assay

NB cell and BMSC viabilities were determined by an Enhancing Cell Counting Kit 8 (CCK-8, Beyotime). In brief, cells (5000 per well) were seeded in 96-well plates. Then, 100 μL medium containing 10 μL CCK-8 solution was added to each well, followed by incubation at 37 °C for 3 h. Next, absorbance was measured at 450 nm.

### 2.8. Cell Treatments

For erastin (4 μM, MCE) and RSL3 (2 μM, MCE) treatments, 90% confluent cells were washed with PBS and cultured in fresh medium. Then, erastin was added to the medium for another 24 h, whereas RSL3 was added for 8 h. Ferrostatin-1 (Fer-1, MCE) was used to treat cells to inhibit ferroptosis. Cisplatin (Beyotime) and H_2_O_2_ (Sigma-Aldrich) were used to treat BMSCs and NB cells for 24 h. Doxorubicin (MCE), etoposide (MCE), and vincristine (MCE) were also used to treat BMSCs and NB cells.

### 2.9. Malondialdehyde (MDA) and GSH Measurements

BMSCs were lysed with cell lysis buffer and the supernatant was used for measurement with a MDA Assay kit (Beyotime) in accordance with the manufacturer’s instructions. To assess the GSH level, cells from each group were collected followed by measure of the GSH level with a GSH and GSSG Assay kit (Beyotime) as the kit instruction.

### 2.10. Glutathione Peroxidases (GPx) Activity Assay and Phosphofructokinase (PFK) Activity Assay

BMSCs were washed with PBS and lysis buffer was used to lyse cells. The supernatant was collected and GPx activity was detected by a Cellular Glutathione Peroxidase Assay Kit (Beyotime) in accordance with the manufacturer’s instructions. To detect the PFK activity, ultrasound was used to disrupt cells and the supernatant was used for measurement with a PFK Activity Assay Kit (Solarbio, Beijing, China).

### 2.11. Western Blot Analysis

RIPA lysis buffer containing a cocktail (50×) was used to lyse cells. After isolating proteins from homogenates, SDS-PAGE gels (Beyotime) were used for electrophoresis. Proteins were transferred onto a PVDF membrane and incubated in 5% dry non-fat milk. Immunoblotting was performed overnight at 4 °C using primary antibodies against SLC7A11 (26864-1-AP, Proteintech, Rosemont, IL, USA), NFS1 (15370-1-AP, Proteintech), GPX4 (ab125066, abcam, Cambridge, UK), Ferritin heavy chain (FTH, ab75972, abcam), Ferritin light chain (FTL, ab75973, abcam), SLC40A1 (FPN, ab239583, abcam), Mitochondrial ferritin (MTFT, ab124889, abcam), Transferrin receptor (TFR, ab214039, abcam), SLC11A2 (divalent metal transporter 1, DMT1, 20507-1-AP, Proteintech), NRF2 (16396-1-AP, Proteintech), and GAPDH (60004-1-Ig, Proteintech). The primary antibodies were diluted in Primary Antibody Dilution Buffer (Beyotime). Horseradish peroxidase-conjugated secondary antibodies (Proteintech) diluted in 5% FBS were then applied to the membranes for another 1.5 h at room temperature. Chemiluminescent signals were detected by the ChemiDoc imaging system (Tanon, Urumqi, China) after development by enhanced chemiluminescence (Millipore, Burlington, MA, USA).

### 2.12. JC-1 and ROS Assays

An enhanced mitochondrial membrane potential assay kit with JC-1 (Beyotime) was used to examine BMSCs treated with H_2_O_2_ and cisplatin. A Reactive Oxygen Species (ROS) Assay Kit (Beyotime) was used to detect the antioxidative capacity of cells. Both JC-1 and ROS assays were observed under a fluorescence microscope.

### 2.13. RNA-Sequencing (RNA-Seq) and Analysis

Total RNA extraction from BMSCs and NB BMSCs cells and cDNA library construction and sequencing were carried out by RNA-seq technology at MingMa Technologies Company (Shanghai, China). The mRNA-focused sequencing libraries from total RNA were prepared using VAHTS mRNA-seq v3 Library Prep Kit (VAHTS, NR611). The sequencing library was examined by Qubit and Agilent to guarantee quality. A NovaSeq 6000 Sequencer in the Illumina platform was used to conduct sequencing in PE150 mode. Fragments per kilobase of exon per million fragments (FPKM) were mapped and significant changes in gene expression were calculated using edgeR (v3.28.1). Gene Ontology and Kyoto Encyclopedia of Genes and Genome were used to assess pathways.

### 2.14. Immunofluorescence Analysis

Paraffin-embedded sections of BM from NB patients with or without BM metastasis were dewaxed. The sections were hydrated and incubated with hydrogen peroxide blocking solution. After antigen retrieval, 5% bovine serum albumin was applied to the sample for 20 min. The sections were incubated with a primary antibody at 4 °C overnight, followed by the secondary antibody for 1 h at room temperature. Tyramide Reagent was applied to samples followed by repetition of another primary antibody. Antibodies against markers used to analyze BMSCs were CD90 and CD105. After mounting, images were captured under a fluorescence microscope.

### 2.15. Animal Experiments

An SK-N-BE(2) cell suspension (1 × 10^6^ cells/100 μL) was injected subcutaneously with or without BMSCs (2  ×  10^5^) into 4-week-old BALB/c nude mice (Jihui Technology Co., China) [28]. The mice were randomly divided into four groups (5 mice/group): (1) control group injected with NB cells only, (2) NB+BMSCs group, (3) erastin group, and (4) RSL3 group. Six days after inoculation, both RSL3 (5 mg/kg) and erastin (10 mg/kg) were administered by intraperitoneal injection in 100 μL once per day [29,30]. After 21 days of treatment, the mice were euthanized. The tumor volume was calculated as 0.5 × length × width^2^ [31]. Animal experiments were approved by the Animal Care and Use Committee, Shanghai Children’s Medical Center, Shanghai Jiao Tong University School of Medicine (China) (No.SCMC-LAWEC-2022-002).

### 2.16. Statistical Analysis

Statistics were performed with GraphPad Prism 8.2.1. Data from at least three repeated experiments (*n* = 3) are expressed as mean values ± standard deviation (SD). Statistically significant differences were calculated by analysis of variance or Student’s *t*-test. *p* < 0.05 was considered statistically significant.

## 3. Results

### 3.1. NB-BMSCs Increased NB Engraftment and Growth

To verify the effects of NB-BMSCs on NB cells, we treated NB cell lines with CM. Cells incubated with CM of NB-BMSCs had significantly improved viability (Figure 1A). Colony formation assays showed that tumor proliferation increased significantly after being treated with CM from NB-BMSCs (Figure 1B,C). Transwell and scratch assays suggested that the migration rates of NB cells significantly increased after NB-BMSC cm treatment (Figure 1D–G). These results show that incubation with NB-BMSC-CMs could promote the growth and migration of NB cells.

### 3.2. NB-BMSCs Were More Resistant to Chemotherapeutic Agents Than Normal BMSCs

To examine the influence of chemotherapeutic agents used in NB patients on BMSCs and NB-BMSCs, cells were treated with cisplatin, etoposide, vincristine, or doxorubicin. The results showed that the concentration of cisplatin that could induce significant apoptosis levels in NB cells had no significant effect on BMSCs or NB-BMSCs (Figure 2A,B). With this same concentration, the apoptosis rates of BMSCs were higher than those of NB-BMSCs after cisplatin treatment (Figure 2C,D). The concentration of etoposide or vincristine that could sufficiently kill the NB cells induced apoptosis in only about 7–22% of BMSCs or NB-BMSCs (Figure 2E–G). With 20 μM doxorubicin, cell viability decreased by less than 10% in BMSCs and NB-BMSCs, while this concentration could kill NB cells thoroughly (Figure 2H). The data also suggested that all the chemotherapy drugs tested could induce higher apoptotic rates in BMSCs than in NB-BMSCs.

### 3.3. NB-CM Treatment Suppressed Anti-Ferroptosis Capacity of BMSCs

To explore approaches to effectively kill NB-BMSCs, we performed RNA-seq to assess gene expression changes in BMSCs induced by NB-CM. The three major mechanisms of ferroptosis resistance, the antioxidant regulator NRF2, SLC7A11/GSH/GPX4 pathway, and mechanistic target of the rapamycin (mTOR) pathway, were all decreased in NB-BMSCs (Figure 3A,B and Appendix A). The protein levels of NRF2, SLC7A11, and GPX4 were also significantly downregulated in NB-BMSCs compared with normal BMSCs (Figure 3C,D). Detecting the GPx activity reveals the overall intracellular antioxidant levels, including the elimination of lipid peroxidation [32,33]. It was found that NB-BMSCs had significantly decreased GPx activity, which indicated lower antioxidant capacity in NB-BMSCs (Figure 3E). These results further indicated a need to explore the mechanism and seek effective ways to kill NB-BMSCs through inducing ferroptosis.

### 3.4. NB-CM Treatment Decreased the Labile Iron Pool (LIP) Levels in BMSCs

To investigate the changes in the anti-ferroptosis system of BMSCs induced by NB cells, we first examined the iron levels, finding that NB-BMSCs had higher iron content (Figure 4A). Iron storage is important for preventing oxidative stress in cells. In this context, we examined the expression levels of three iron storage proteins [34]. FTH, FTL, and ferritin in mitochondria (FTMT) protein levels were significantly lower in NB-BMSCs. TFRC and DMT1, which are involved in iron import, had similar expression levels in NB-BMSCs and normal BMSCs, while FPN1, which transports cellular iron out of cells, had significantly decreased protein levels in NB-BMSCs (Figure 4B–H) [35,36,37]. These Western blotting results were consistent with the RNA-seq results (Appendix A). We also observed decreased lipid peroxidation in NB-BMSCs (Figure 4I,J). These results indicate that the levels of LIP and ferritin were lower in NB-BMSCs compared with those in normal BMSCs.

We further explored the application of iron in NB-BMSCs. Therefore, we analyzed NFS1, the main protein that participates in synthesizing ISCs [38]. Our results showed that NFS1 was significantly increased in NB-BMSCs (Figure 4K,L). We further found that the heme levels in NB-BMSCs were significantly increased (Figure 4M). These results suggested that NB-BMSCs mainly used iron to synthesize heme and ISCs, while cellular ferritin and free iron levels were lower. We also used H_2_O_2_ treatment to examine the LIP levels in NB-BMSCs. NB-BMSCs had higher cell viability rates and lower intracellular ROS levels than normal BMSCs following H_2_O_2_ treatment (Figure 4N–Q). JC-1 assay indicated higher rate of abnormal mitochondrial membrane potential in BMSCs treated with H_2_O_2_ (Figure 4R,S). We also measured the ATP levels, which can reflect the metabolic status of cells. NB-BMSCs had rapidly increased ATP levels that significantly exceeded those of normal BMSCs after H_2_O_2_ treatment. To test whether this phenomenon was caused by different metabolic pathway, we detected phosphofructokinase—the key enzyme in glycolysis—and found that the difference in ATP levels might be due to the enhanced degree of glycolysis in NB-BMSCs (Figure 4T and Appendix A). These results further demonstrate that LIP levels were decreased in NB-BMSCs, which are different from tumor cells.

### 3.5. NB-BMSCs Were Sensitive to Ferroptosis Inducers

To examine the effect of ferroptosis inducers on BMSCs and NB-BMSCs, we treated these cells with erastin or RSL3. Both of the inducers could significantly induce ferroptosis in NB cells (Figure 5A). NB-BMSCs had lower GSH levels compared to normal BMSCs (Figure 5B). The IC50 value of NB-BMSCs treated with erastin or RSL3 was lower than that of BMSCs (Figure 5C,D and Appendix A). Both treatments could significantly induce ferroptosis, which included increased lipid peroxidation levels and decreased cell viability rates in NB-BMSCs. Although erastin and RSL3 affected normal BMSCs, the ferroptosis levels were higher in NB-BMSCs than in BMSCs (Figure 5E–J). Crystal violet staining also revealed that normal BMSCs had significantly lower cell death rates compared with NB-BMSCs after erastin or RSL3 treatment (Figure 5K). Ferroptosis inhibitor Fer-1 treatment significantly inhibited the effects of erastin or RSL3. Our results suggest that these ferroptosis inducers could effectively kill NB-BMSCs while exerting little effect on normal BMSCs.

### 3.6. Ferroptosis Induction Was Effective in CAF-MSCs from NB Patients and an NB Mouse Model

To systematically identify the impact of ferroptosis inducers on BMSCs affected by NB cells, we isolated and characterized CAF-MSCs from NB patient tumors. These CAF-MSCs had an extremely similar phenotype to NB-BMSCs (Figure 6A,B and Appendix A). Tumors can induce BMSCs to migrate into the microenvironment and educate them, which were the origin of CAF-MSCs. CAF-MSCs in tumor tissues are directly affected by NB cells. We treated CAF-MSCs with the same concentration of cisplatin used in the previous experiments, finding that this drug could not induce significant levels of apoptosis (Figure 6C). Treatment with etoposide, vincristine, and doxorubicin also showed similar effects on these CAF-MSCs as on NB-BMSCs (Appendix A). H_2_O_2_ treatment could decrease CAF-MSC viability by 8% to 15%, which was similar to rates seen in NB-BMSCs (Figure 6D). Next, we applied ferroptosis inducers erastin and RSL3, which both could induce severe ferroptosis in CAF-MSCs at concentrations similar to those used in NB-BMSCs (Figure 6E–H). We then used a mouse model to examine our hypothesis that NB-BMSCs could promote NB growth while ferroptosis inducers could inhibit NB-BMSCs. Neither of the ferroptosis inducers significantly affected body weight, blood routine, or blood biochemical indexes (Appendix A). Our results indicated that injecting mice with NB cells together with BMSCs significantly promoted tumor progression compared with injecting NB cells only. We also found that both erastin and RSL3 effectively inhibited tumors originating from NB and BMSCs (Figure 6I–K).

### 3.7. BM from NB Patients with Metastasis Had Lower SLC7A11 and GPX4 Levels

To continue our work in vivo, we collected BM from NB patients with or without BM metastasis. In accordance with previous studies that identified BMSCs in NB-BM, we performed double staining of CD90 and CD105 as markers of BMSCs. The results showed that SLC7A11 expression levels in BMSCs of patients with BM metastasis were significantly lower than those in patients without BM metastasis (Figure 7A,B and Appendix A). GPX4 expression levels were also decreased in patients with BM metastasis (Figure 7C,D and Appendix A). We then measured NFS1 expression levels, finding them to be significantly higher in children with BM metastasis than in those without metastasis (Figure 7E,F and Appendix A). These results confirmed our in vitro data, indicating that BMSCs had a decreased anti-ferroptosis capacity when influenced by NB.

## 4. Discussion

NB accounts for 8% to 10% of malignant tumors in children, and distant metastasis is the major indicator of high-risk NB. BM is the most common distant metastasis site of NB, accounting for 60%. Despite intensive multimodal therapy, the survival rate of NB patients with BM dissemination remains less than 50% [1,2]. In accordance with the “seed and soil” theory of tumor metastasis, tumors tend to colonize and grow in suitable microenvironments [39,40,41]. BM can provide a favorable site for NB growth. Recent studies have found that tumors can affect cells in the microenvironment, such as MSCs, to make it a conducive place for growth and metastasis [9,42,43]. BMSCs have also been studied in NB, and their NB-induced production of certain cytokines can promote tumor growth [11,44]. These cells play a major role in promoting tumor progression in NB patients with BM metastasis. However, there is currently no effective method to control BMSCs in tumors. These tumor-associated BMSCs are resistant to the effects induced by chemotherapeutic drugs and can secrete specific cytokines that help protect tumors [14,15]. Therefore, new approaches need to be identified to treat BMSCs in NB patients with BM metastasis.

We first used the CM of NB-BMSCs to treat NB cells. The results showed that NB-BMSCs could significantly promote the proliferation and migration of NB cell lines, consistent with previous studies. Next, we focused on various treatments of NB-BMSCs. We used cisplatin, etoposide, vincristine, and doxorubicin, which are the most commonly used drugs in high-risk NB patients. However, we observed little effect. The effective concentration of each drug that could induce severe apoptosis levels in NB cells had little effect on BMSCs, especially NB-BMSCs. Our results suggest that a considerable number of tumor-associated BMSCs could likely survive in the tumor microenvironment following chemotherapeutic treatment. Therefore, classical chemotherapeutic treatments did not result in significant killing of NB-BMSCs, and focus should be given to other methods of therapeutic intervention.

As a novel form of cell death, ferroptosis has attracted wide attention in tumor therapy [45]. Recent studies have focused on using drugs, gene editing techniques, or nanomaterials to induce ferroptosis through various mechanisms to effectively kill cancer cells [46,47,48,49]. We performed an RNA-seq analysis of normal BMSCs and NB-BMSCs to identify differentially expressed genes that correlated with ferroptosis. We found that three major anti-ferroptosis mechanisms, NRF2, the SLC7A11/GSH/GPX4 pathway, and the mTOR pathway, were all decreased in NB-BMSCs [19]. Our findings suggest that NB-BMSCs might be vulnerable to these classic methods of inducing ferroptosis.

Currently, there are three methods that have been widely recognized to induce ferroptosis in tumors: inhibition of system Xc^−^ or GPX4, inactivation of CoQ10 reductases, and induction of lipid peroxidation such as targeting free iron in the LIP [22,50,51,52]. RNA-seq analysis showed no significant difference in CoQ10 reductase (AIFM2 and DHODH) mRNA levels between BMSCs and NB-BMSCs. LIP is an intracellular nonprotein bound iron that can generate peroxidation of membrane-bound iron via the Fenton reaction, resulting in oxidative cell damage [53,54,55]. We examined levels of iron metabolism, finding that NB-BMSCs had lower levels of LIP that maintained iron homeostasis with decreased anti-ferroptosis capacity. We first found that ferritin and FPN expression levels were both downregulated in NB-BMSCs. Ferritin, which includes FTH, FTL, and FTMT, is used to store iron and can help protect cells from oxidative stress caused by an overload of redox active free iron [34,56]. FTH and FTMT serve as ferroxidases to allow storage of ferric hydroxides (Fe^3+^) instead of reactive ferrous iron (Fe^2+^) into FTL. FPN is the only protein that transfers iron out of the cell. Decreased levels of cellular ferritin and FPN were corelated with low iron content [35]. However, we detected higher iron content in NB-BMSCs, which seemed contradictory. We speculated that the iron in NB-BMSCs was used to synthesize iron-related compounds other than ferritin.

Aside for ferritin, iron is mainly used to synthesize ISCs and heme. ISCs exist in mitochondria and are related to electron transport, enzyme activity regulation, and catalytic functions [57]. As important cofactors of redox and iron homeostasis, ISCs and their associated regulatory pathways enable cells to escape ferroptosis by reducing LIP [58]. Cysteine desulphurase NFS1 is involved in ISC biosynthesis by collecting sulfur from cysteine. It can promote tolerance against cellular stressors and regulate iron homeostasis [38,59]. Here, we found that NFS1 expression levels were higher in NB-BMSCs. Heme plays important roles in major cellular processes, such as oxygen transport, respiratory complexes, electron transfer chains, and storage of metal ions [60]. We found that heme levels were higher in NB-BMSCs, as expected. Increased amounts of heme could support high NB-BMSCs viability in a complicated environment. Higher levels of both NFS1 and heme suggested that NB-BMSCs used most of the available iron to synthesize iron-compounds to withstand the complex tumor microenvironment. This also possibly explains the stronger resistance of NB-BMSCs to chemotherapeutic treatments. These results were also confirmed by lower lipid peroxidation levels and reduced sensitivity to H_2_O_2_, which indicated decreased free iron in NB-BMSCs.

Therefore, we used classic inhibitors erastin and RSL3, which target the Xc^−^/GSH/GPX4 pathway [23,24]. Both achieved satisfactory results by killing numerous NB-BMSCs, while having little effect on normal BMSCs. These results confirmed that ferroptosis inducers were extremely efficient in killing NB-BMSCs, while ferroptosis might be an advisable way to clear tumor-associated BMSCs. We also extracted CAF-MSCs from tumors of NB patients to examine the MSCs that were directly influenced by the tumor. Previous studies found that BMSCs can be recruited by tumors and transformed into tumor associated MSCs or CAFs [61,62,63]. These cells, referred to as CAF-MSCs, can secrete various cytokines to promote tumor progression. In this study, the observed CAF-MSC phenotype was similar to that of NB-BMSCs. These CAF-MSCs also showed weak resistance to classic ferroptosis inducers, which further confirmed that targeting anti-ferroptosis pathways is the optimal choice for treating tumor-associated MSCs. We analyzed the BM from NB patients and found that BMSCs of patients with BM metastasis had lower anti-ferroptosis levels by detecting SLC7A11 and GPX4 expression. They also had higher NFS1 expression compared with NB patients without metastasis, trends that were consistent with our in vitro results. Moreover, we identified that BMSCs could significantly promote NB growth, while erastin and RSL3 could reverse this phenomenon in a mouse model. This further explained the major role of BMSCs in promoting tumor progression and the effectiveness of pro-ferroptosis treatment. Our multifaceted research provides strong evidence supporting the effectiveness and safety of targeting anti-ferroptosis treatment in NB patients, even in patients with BM metastasis.

In this study, we examined the effects of NB on BMSC. We experimentally identified a method to sufficiently kill them, while having little effect on normal BMSCs. However, this study has several limitations. First, we did not directly detect ISCs because there is no mainstream method available to measure its content. Therefore, we analyzed NFS1 levels to represent ISC synthesis. Although it has been reported that several chemotherapeutic drugs, such as cisplatin, commonly used in NB patients, could partly induce ferroptosis, the results of promoting ferroptosis to these drugs remains unknown in NB patients. Moreover, no pro-ferroptotic drugs have been used in clinical translation of NB. Further research will be performed to explore the drugs which can promote ferroptosis in NB patients to potentially achieve clinical application as soon as possible.

## 5. Conclusions

Here, we explored several changes related to iron metabolism in BMSCs induced by NB which play an important role in tumor progression. Our data suggest that targeting an anti-ferroptosis system, especially the Xc^−^/GSH/GPX4 pathway, was an effective method to kill NB-BMSCs. Both erastin and RSL3 could effectively kill NB-BMSCs with little effect on normal BMSCs. Our study provides a new strategy to treat NB patients through clearance of tumor-associated BMSCs, which can help inhibit tumor recolonization and growth effectively.

## Figures and Tables

**Figure 1 cancers-15-01301-f001:**
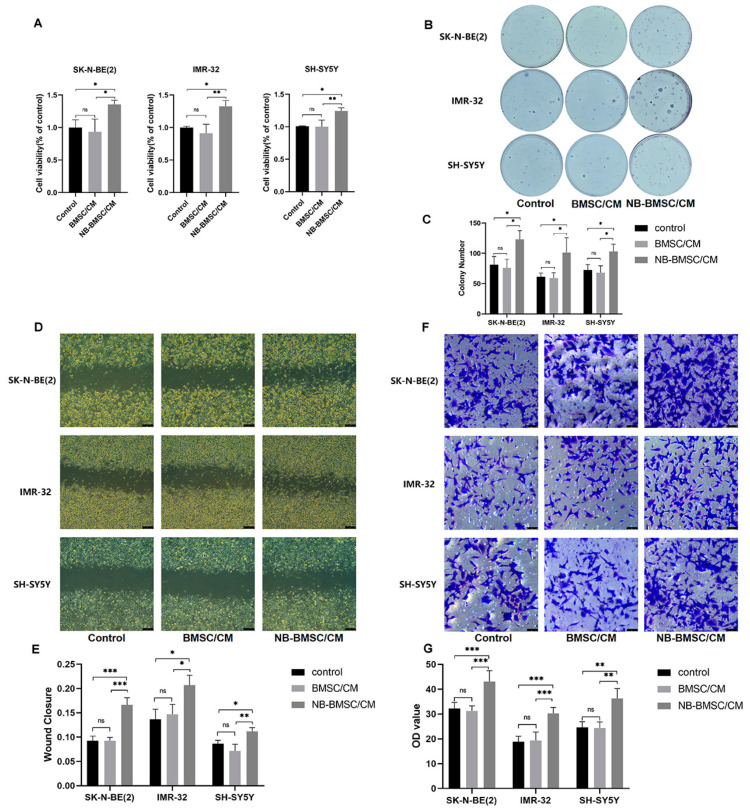
NB-BMSCs promoted NB growth and migration. (**A**) CCK8 assay was performed to assess NB cell viability. (**B**,**C**) Colony formation assay was performed to assess the proliferation of NB cell. (**D**–**G**) Scratch assay and transwell assay was performed to examine the migration ability of NB cells. Scale bar (**D**): 250 μm; Scale bar (**F**): 50 μm. A one-way analysis of variance was used. ^ns^ not significant; * *p* < 0.05; ** *p* < 0.01; *** *p* < 0.001.

**Figure 2 cancers-15-01301-f002:**
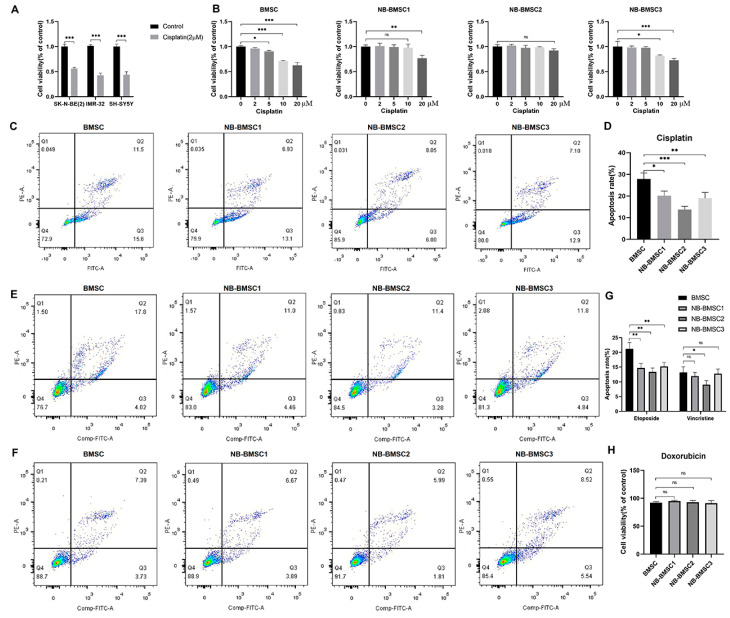
Chemotherapeutic drugs cannot kill NB-BMSCs effectively. (**A**) CCK8 assay was performed to assess NB cell viability under 2 μM cisplatin. (Student’s *t*-test was used). (**B**) Cell viability was assessed after 0–20 μM cisplatin was used to treat BMSCs and NB-BMSCs. (**C**,**D**) 10 μM cisplatin was used to assess the anti-cisplatin ability of normal BMSCs and NB-BMSCs. (**E**–**G**) Apoptosis rate of normal BMSCs and NB-BMSCs was examined under 10 μM etoposide or vincristine (Most of NB cells were in apoptosis under this concentration). (**H**) Cell viability of normal BMSCs and NB-BMSCs under 20 μM doxorubicin (NB cells were all in apoptosis under this concentration). A one-way analysis of variance was used. ^ns^ not significant; * *p* < 0.05; ** *p* < 0.01; *** *p* < 0.001.

**Figure 3 cancers-15-01301-f003:**
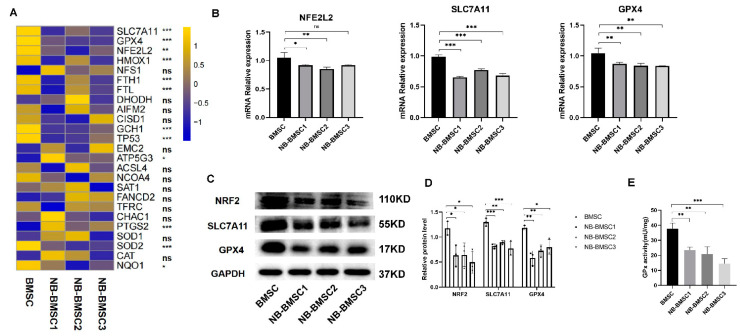
RNA-seq was performed and anti-ferroptosis pathways were selected. (**A**,**B**) The pathways linked with ferroptosis were selected. (**C**,**D**) Western blotting was used to analyze protein levels of NRF2, SLC7A11 and GPX4. (**E**) GPx activity was detected in BMSCs and NB-BMSCs. A one-way analysis of variance was used. ^ns^ not significant; * *p* < 0.05; ** *p* < 0.01; *** *p* < 0.001.

**Figure 4 cancers-15-01301-f004:**
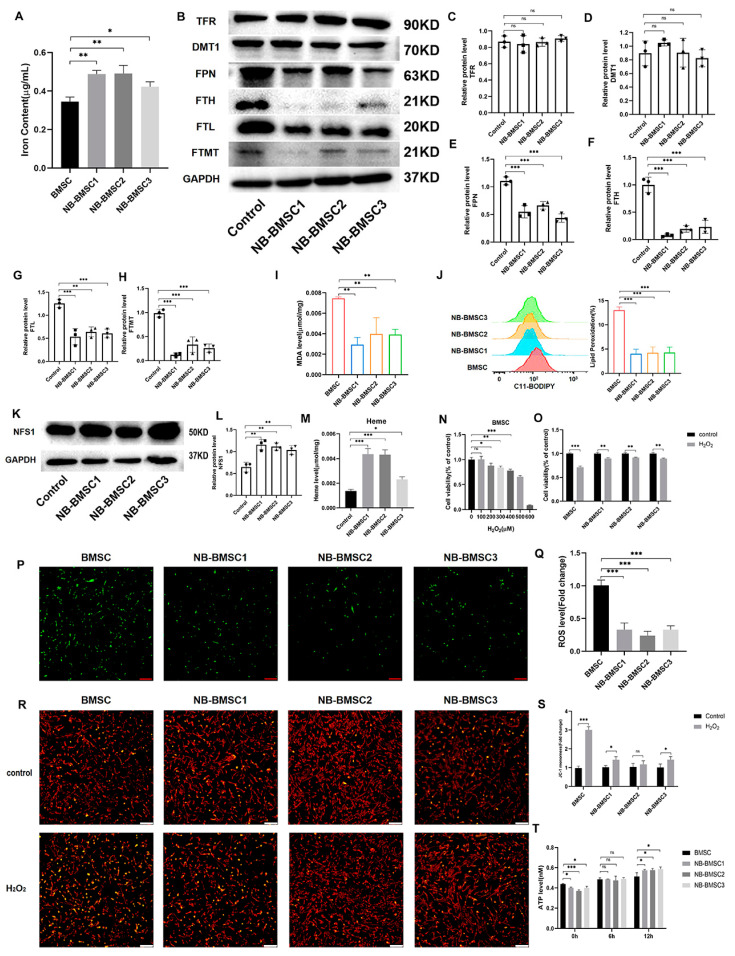
NB-BMSCs had smaller LIP. (**A**) Iron content of normal BMSCs and NB-BMSCs was detected. (**B**–**H**) Western blotting was used to analyze protein levels of TFR, DMT1, FPN, FTH, FTL, FTMT. (**I**,**J**) MDA level and lipid peroxidation level was used to reflect the ferroptosis level of normal BMSCs and NB-BMSCs. (**K**,**L**) NFS1 level was detected by Western blotting. (**M**) A heme assay kit was used to detect the heme level of normal BMSCs and NB-BMSCs. (**N**) Cell viability of BMSCs were assessed after 0–600 μM H_2_O_2_ treatment, and (**O**) 400 μM H_2_O_2_ were used to assess the cell viability of normal BMSCs and NB-BMSCs. (**P**,**Q**) ROS level was detected in normal BMSCs and NB-BMSCs. Scale bar: 250 μm. (**R**,**S**) JC-1 assay was performed to assess the changes in mitochondrial membrane potential. Scale bar: 250 μm. (**T**) ATP level was used to reflect the metabolic status of cells. Student’s *t*-test was used in analysis of (**O**,**S**), while a one-way analysis of variance was used in other analysis. ^ns^ not significant; * *p* < 0.05; ** *p* < 0.01; *** *p* < 0.001.

**Figure 5 cancers-15-01301-f005:**
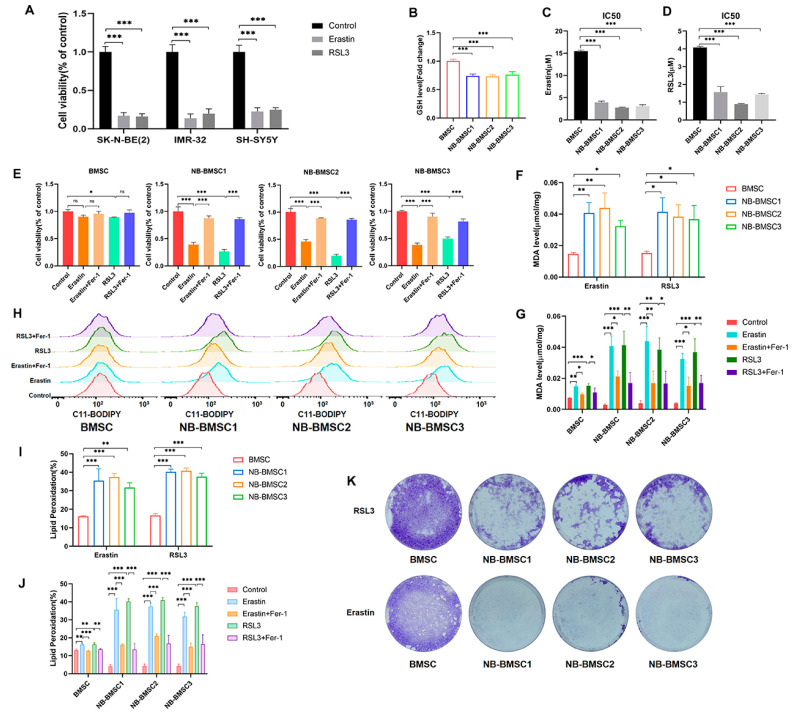
NB-BMSCs had lower anti-ferroptosis capacity. (**A**) CCK8 assay was performed to assess NB cell viability under 4 μM erastin and 2 μM RSL3 treatment. (**B**) GSH level of normal BMSCs and NB-BMSCs was examined. (**C**,**D**) We detected the IC50 of erastin and RSL3 on normal BMSCs and NB-BMSCs. (**E**–**J**) 4 μM erastin and 2 μM RSL3 was used to treat normal BMSCs and NB-BMSCs; Cell viability, MDA level, and lipid peroxidation level was detected to reflect the ferroptosis level. (**K**) Crystal violet staining was used to indicate condition of normal BMSCs and NB-BMSCs under 10 μM erastin and 2 μM RSL3. A one-way analysis of variance was used. ^ns^ not significant; * *p* < 0.05; ** *p* < 0.01; *** *p* < 0.001.

**Figure 6 cancers-15-01301-f006:**
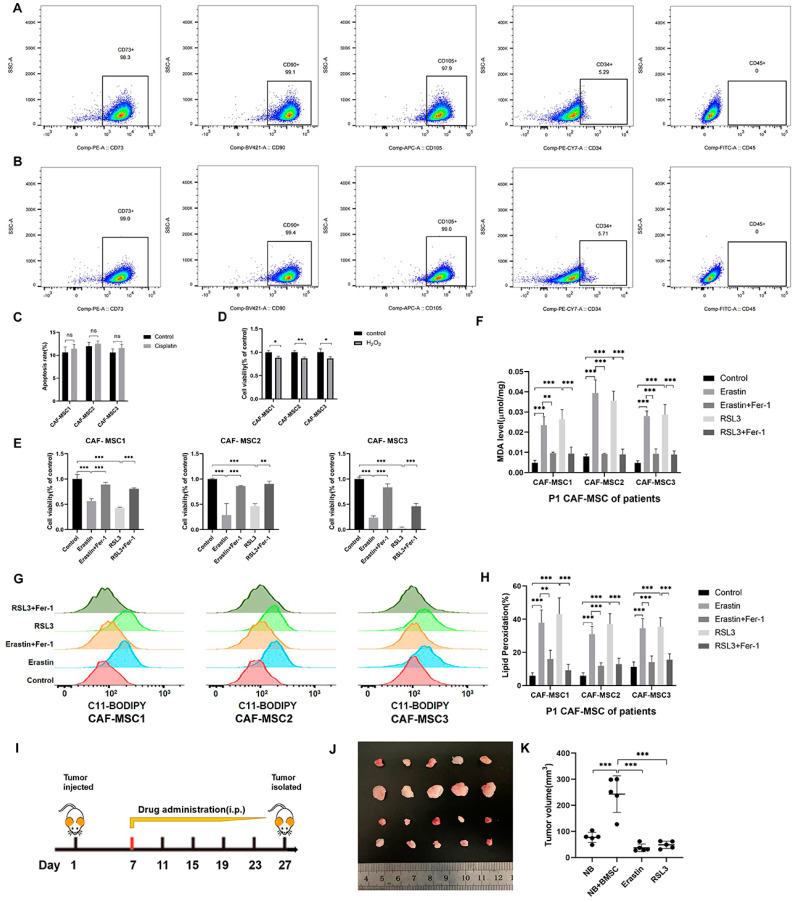
CAF-MSCs of NB patients and mouse model supported our viewpoint. (**A**,**B**) FACS analysis for detecting surface markers of NB-BMSCs and CAF-MSCs of NB patients. (**C**) Apoptosis rate of CAF-MSCs from NB patients was examined under 10 μM cisplatin. (**D**) Cell viability of CAF-MSCs was detected under 400 μM H_2_O_2_. (**E**–**H**) Cell viability, MDA level, and lipid peroxidation level were detected to reflect the ferroptosis level under 4 μM erastin and 2 μM RSL3. (**I**–**K**) Mouse modelling was conducted and tumor progression was observed. Student’s *t*-test was used in analysis of (**C**,**D**), while a one-way analysis of variance was used in other analysis. ^ns^ not significant; * *p* < 0.05; ** *p* < 0.01; *** *p* < 0.001.

**Figure 7 cancers-15-01301-f007:**
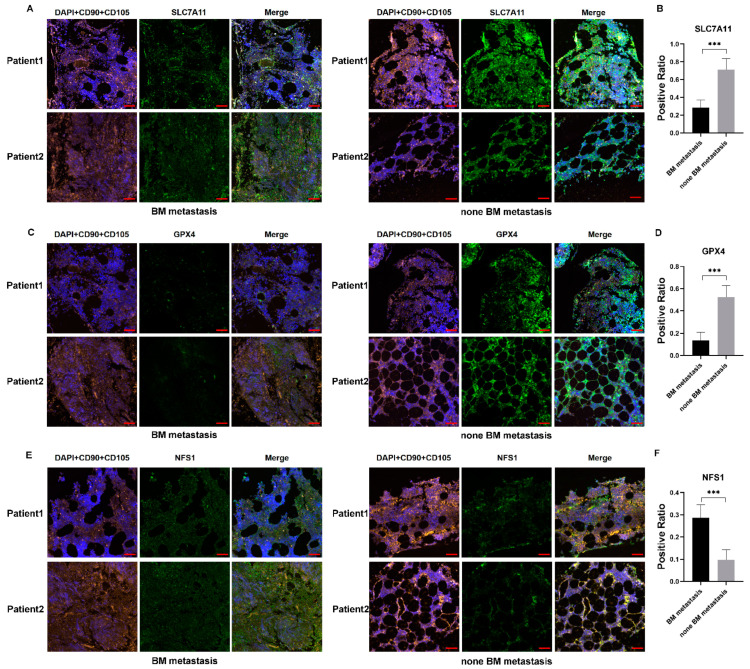
Immunofluorescence indicated decreased anti-ferroptosis capacity of BMSCs in patients with BM metastasis. (**A**–**F**) CD90 and CD105 were used to mark BMSCs; SLC7A11, GPX4, and NFS1 were detected in BM of NB patients with or without BM metastasis. Scale bar: 50 μm. Student’s *t*-test was used in the analysis. *** *p* < 0.001.

## Data Availability

All data generated or analyzed during this study are included in this manuscript.

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
