# Peer review of "Ferroptosis Inducers Kill Mesenchymal Stem Cells Affected by Neuroblastoma"

_cancers, 2023, doi:10.3390/cancers15041301_

Round 1

Reviewer 1 Report

In this Paper  Li et al. Described  that  inducing ferroptosis could be a new potential  therapeutic approaches for killing  NB-BMSCs.The paper is not well written, the experiments are quite clear, but not well described.  This reviewer has few major points to ask:

-The authors have to explain better cells model system that they use in the study. For example it is not clear which cells are NB-BMSCs?

-The author have to initiate all paragraph of the results explaining why the decide to do the experiment proposed. Moreover, they have to describe experiments using scientific language. For example: in line 233 “a little apoptosis” they have to indicate the percentage, little is not a scientific term, in line 235 “more apoptosis”, they have to indicate how much in percentage, and so on through the paper.

- In all figure’s captions it is not clear the statistical analysis used.

-In figure 7 the authors have to show a quantification of immunofluorescence analysis

Author Response

In this Paper, Li et al. Described  that  inducing ferroptosis could be a new potential  therapeutic approaches for killing NB-BMSCs. The paper is not well written, the experiments are quite clear, but not well described. 

The author’s response: Thank you very much for your comment. We have improved the description of our results and polished up our paper.

1. The authors have to explain better cells model system that they use in the study. For example it is not clear which cells are NB-BMSCs?

The author’s response: Thanks for your suggestion. BMSCs cultured with conditioned medium of NB cells were named NB-BMSCs. We have redescribed the sentence as: “We used CM of different NB cells to culture BMSCs. BMSCs cultured with CM of SK-N-BE(2), IMR-32, and SH-SY5Y cells were named NB-BMSC1, NB-BMSC2, and NB-BMSC3 cells, respectively”. (Please see page 3, line 107-109)

2. The author have to initiate all paragraph of the results explaining why the decide to do the experiment proposed. Moreover, they have to describe experiments using scientific language. For example: in line 233 “a little apoptosis” they have to indicate the percentage, little is not a scientific term, in line 235 “more apoptosis”, they have to indicate how much in percentage, and so on through the paper.

The author’s response: Thank you very much for your comments. Firstly, we have added experimental purpose in this manuscript as follows: “To verify the effects of NB-BMSCs on NB cells, we treated NB cell lines with CM…” in result 3.1, “To examine the influence of chemotherapeutic agents used in NB patients on BMSCs and NB-BMSCs, cells were treated with …” in result 3.2, and so on. (Please see page 5, line 232; page 6, line245-246)

Secondly, we have carefully checked the manuscript and rewritten all of the unscientific language. For example: “Etoposide or vincristine in the concentration of killing the NB cells sufficiently induced a little apoptosis on BMSCs or NB-BMSCs” was redescribed as “The concentration of etoposide or vincristine that could sufficiently kill the NB cells induced apoptosis in only about 7%- 22% of BMSCs or NB-BMSCs”. (Please see page 7, line 250-252)

3. In all figure’s captions it is not clear the statistical analysis used.

The author’s response: Thank you very much for your suggestion. We have added all the statistical analysis used in the figure’s captions, such as “One-way analysis of variance was used. * p < 0.05; ** p < 0.01; *** p < 0.001”. (Please see page 6, line 243)

4. In figure 7 the authors have to show a quantification of immunofluorescence analysis.

The author’s response: Thank you very much for your timely reminder. We have shown the quantification of GPX4, SLC7A11 and NFS1 expression detected by immunofluorescence. (Please see page 13, Figure 7)

Reviewer 2 Report

In the paper entitled “Ferroptosis inducers kill mesenchymal stem cells affected by neuroblastoma”, Li and co-authors have investigated the therapeutic potential  of ferroptosis in tumor-associated BMSCs of Neuroblastoma.

The paper is well organized and the topics is very interesting. However, in order to better support the results, additional experiments are necessary.

In order to help the authors to improve the article, here below are the main critical points under question that need to be addressed:

Major points:

-          Lines 247-251: Due to the crucial role of GPX4 in ferroptosis, it is necessary to evaluate also its activity.

-          Lines 276-278: The high levels of ATP found in NB-BMSCs is the results of different metabolic pathway used by these cells and BMSCs? The analysis of OXPHOS and glycolysis is required.

-          Lines 323-326: Which are the effects of Erastin and RSL3 on the other mouse organs and on hematic parameters?

-          Lines 340-342: Since the ferroptotic death is strictly related to the intracellular GSH levels, is it possible to measure GSH levels and/or the activity of GSH-related enzymes in patients'samples?

Lines 449-452: The authors concluded affirming that theire study provides a new strategy to treat NB patients. Have the authors informations about drugs commonly used in clinic that could exert pro-ferroptotic actions and that caould be used in NB patients?

Author Response

In the paper entitled “Ferroptosis inducers kill mesenchymal stem cells affected by neuroblastoma”, Li and co-authors have investigated the therapeutic potential  of ferroptosis in tumor-associated BMSCs of Neuroblastoma. The paper is well organized and the topics is very interesting. However, in order to better support the results, additional experiments are necessary. In order to help the authors to improve the article, here below are the main critical points under question that need to be addressed:

1. Lines 247-251: Due to the crucial role of GPX4 in ferroptosis, it is necessary to evaluate also its activity.

The author’s response: Thank you very much for your helpful suggestion. We have supplied the evaluation of the GPX4 activity in the Methods as follows: “BMSCs were washed with PBS and lysis buffer was used to lysed cells. The supernatant was collected and GPX4 activity was detected by a Cellular Glutathione Peroxidase Assay Kit (Beyotime) in accordance with the manufacturer’s instructions”. The result was showed in Figure 3C and described as follows: “We found that NB-BMSCs had significantly decreased GPX4 activity”. (Please see page 4, line 168-171; page 7, line 272; and Figure 3E)

2. Lines 276-278: The high levels of ATP found in NB-BMSCs is the results of different metabolic pathway used by these cells and BMSCs? The analysis of OXPHOS and glycolysis is required.

The author’s response: Thanks for your suggestion. Phosphofructokinase is the key enzyme in OXPHOS and glycolysis. Therefore, we examined the expression of phosphofructokinase to reflect the levels of OXPHOS and glycolysis. The result was described as follows: “NB-BMSCs had rapidly increased ATP levels that significantly exceeded those of normal BMSCs after H2O2 treatment. To test whether this phenomenon was caused by different metabolic pathway, we detected phosphofructokinase—the key enzyme in glycolysis, and found the difference in ATP levels might be due to the enhanced degree of glycolysis in NB-BMSCs”. (Please see page 4, line 171-173; page 8, line 303-306; and Figure S1)

3. Lines 323-326: Which are the effects of Erastin and RSL3 on the other mouse organs and on hematic parameters?

The author’s response: Thank you very much for your reminder. We found no significant differences on body weight, blood routine and blood biochemical indexes after erastin or RSL3 treatment. We showed these results in supplementary data. Moreover, the doses of erastin or RSL3 used in our study were both within the safe limits of existing studies. (Please see page 11, line 356-358; Figure S3; and Ref 1-3)

  1. Chen, H.; Qi, Q.; Wu, N.; Wang, Y.; Feng, Q.; Jin, R.; Jiang, L. Aspirin promotes RSL3-induced ferroptosis by suppressing mTOR/SREBP-1/SCD1-mediated lipogenesis in PIK3CA-mutatnt colorectal cancer. Redox Biol. 2022, 55, 102426.
  2. Bao, Z.; Hua, L.; Ye, Y.; Wang, D.; Li, C.; Xie, Q.; Wakimoto, H.; Gong, Y.; Ji, J. MEF2C silencing downregulates NF2 and E-cadherin and enhances Erastin-induced ferroptosis in meningioma. Neuro Oncol. 2021, 23, 2014-2027.
  3. Li, Y.; Zeng, X.; Lu, D.; Yin, M.; Shan, M.; Gao, Y. Erastin induces ferroptosis via ferroportin-mediated iron accumulation in endometriosis. Hum Reprod. 2021, 36, 951-964.

4. Lines 340-342: Since the ferroptotic death is strictly related to the intracellular GSH levels, is it possible to measure GSH levels and/or the activity of GSH-related enzymes in patients'samples?

The author’s response: Thank you for your comment. Bone marrow tissues were acquired from puncture so that they were really rare. We had consumed all the samples in this study. Moreover, these samples used in our study were freezing tissues, we apologized for not being able to obtain BMSCs from them. We highly approve of your proposal that comparing GSH level in patients’ samples. We would collect BMSCs from fresh tissues in order to compare these indexes.

5. Lines 449-452: The authors concluded affirming that theire study provides a new strategy to treat NB patients. Have the authors informations about drugs commonly used in clinic that could exert pro-ferroptotic actions and that could be used in NB patients?

The author’s response: Thank you very much for your comment. We discovered the effect of classical ferroptosis inducers in killing NB associated BMSCs in this study. Although it has been reported that several chemotherapeutic drugs, such as cisplatin, commonly used in NB patients, could partly induce ferroptosis, the ability of promoting ferroptosis about these drugs remain unknown in NB patients. Moreover, no pro-ferroptotic drugs have been used in clinical translation of NB so that we would further explore drugs which can promote ferroptosis in NB patients. We have added this content in the Discussion section. (Please see page 15, line 478-483)

Round 2

Reviewer 1 Report

This reviewer is satisfied by the author's corrections.  In the present form the paper is acceptable for publication on Cancers

Author Response

This reviewer is satisfied by the author's corrections. In the present form the paper is acceptable for publication on Cancers.

The author’s response: Thank you very much for your recognition to our paper.

Reviewer 2 Report

Dear authors, I really appreciate your efforts to satisfy all my requests and clarify my perplexities. However, I suggest one more small change regarding GPX4 activity. In fact, the kit used to evalaute GPX4 activity is not specific as it is able to evaluate the activity of all GPXs and not just the GPX4. For this reason, it cannot be stated that NB-BMSCs had significantly decreased GPX4 acitivity. Please rewrite this sentence in accordance with this fact.

Author Response

Dear authors, I really appreciate your efforts to satisfy all my requests and clarify my perplexities. However, I suggest one more small change regarding GPX4 activity. In fact, the kit used to evalaute GPX4 activity is not specific as it is able to evaluate the activity of all GPXs and not just the GPX4. For this reason, it cannot be stated that NB-BMSCs had significantly decreased GPX4 acitivity. Please rewrite this sentence in accordance with this fact.

The author’s response: Thank you very much for your recognition and sorry for our mistake. We have rewrited the sentence as follows: “Detecting the glutathione peroxidases (GPx) activity reveals the overall intracellular antioxidant levels, including the elimination of lipid peroxidation. It was found that NB-BMSCs had significantly decreased GPx activity, which indicated lower antioxidant capacity in NB-BMSCs”.( Please see page 8, line272-275)